# LEARNING TO DOWNSAMPLE FOR SEGMENTATION OF ULTRA-HIGH RESOLUTION IMAGES

**Chen Jin**[1], **Ryutaro Tanno**[2], **Thomy Mertzanidou**[1], **Eleftheria Panagiotaki**[1], **Daniel C. Alexander**[1]

[1] Centre for Medical Image Computing, Department of Computer Science, University College London
[2] Healthcare Intelligence, Microsoft Research Cambridge
{chen.jin,t.mertzanidou,e.panagiotaki,d.alexander}@ucl.ac.uk
rytanno@microsoft.com

## ABSTRACT

Many computer vision systems require low-cost segmentation algorithms based on deep learning, either because of the enormous size of input images or limited computational budget. Common solutions uniformly downsample the input images to meet memory constraints, assuming *all pixels are equally informative*. In this work, we demonstrate that this assumption can harm the segmentation performance because the segmentation difficulty varies spatially (see Figure 1 "Uniform"). We combat this problem by introducing a *learnable downsampling module*, which can be optimised together with the given segmentation model in an end-to-end fashion. We formulate the problem of training such downsampling module as optimisation of sampling density distributions over the input images given their low-resolution views. To defend against degenerate solutions (e.g. over-sampling trivial regions like the backgrounds), we propose a regularisation term that encourages the sampling locations to concentrate around the object boundaries. We find the downsampling module learns to sample more densely at difficult locations, thereby improving the segmentation performance (see Figure 1 "Ours"). Our experiments on benchmarks of high-resolution street view, aerial and medical images demonstrate substantial improvements in terms of efficiency-and-accuracy trade-off compared to both uniform downsampling and two recent advanced downsampling techniques.

## 1 INTRODUCTION

Many computer vision applications such as auto-piloting, geospatial analysis and medical image processing rely on semantic segmentation of ultra-high resolution images. Exemplary applications include urban scene analysis with camera array images ($> 25000 \times 14000$ pixels)(Wang et al., 2020), geospatial analysis with satellite images ($> 5000 \times 5000$ pixels) (Maggiori et al., 2017) and histopathological analysis with whole slide images ($> 10,000 \times 10,000$ pixels) (Srinidhi et al., 2019). Computational challenges arise when applying deep learning segmentation techniques on those ultra-high resolution images.

To speed up the performance, meet memory requirements or reduce data transmission latency, standard pipelines often employ a preprocessing step that *uniformly downsamples* both input images and labels, and train the segmentation model at lower resolutions. However, such uniform downsampling operates under an unrealistic assumption that all pixels are equally important, which can lead to under-sampling of salient regions in input images and consequently compromise the segmentation accuracy of trained models. Marin et al. (2019) recently argued that a better downsampling scheme should sample pixels more densely near object boundaries, and introduced a strategy that adapts the sampling locations based on the output of a separate edge-detection model. Such "edge-based" downsampling technique has achieved the state-of-the-art performance in low-cost segmentation benchmarks. However, this method is not optimised directly to maximise the downstream segmentation accuracy; importantly, the pixels near the object boundaries are not necessarily the most informative about their semantic identities; for instance, textual cues indicating forest or cancer regions are less dependent on tree or cell boundaries, respectively.

---

Video demos available at https://lxasqjc.github.io/learn-downsample.github.io/

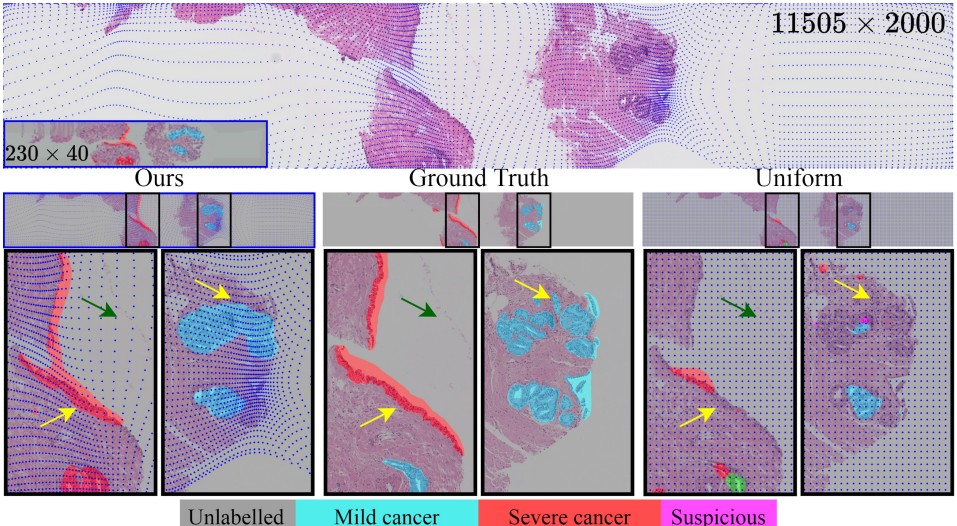

Figure 1: **Semantic segmentation with learnable downsampling operation on cancer histology images.** Top row: Our *deformation module* learns to adapt the sampling locations spatially (blue dots on the images) according to their utility for the downstream segmentation task and produces a deformed downsampled image (bottom-left corner), which leads to a more accurate low-resolution segmentation. Middle row: high-resolution segmentation reversed from low-resolution predictions. Bottom row: Compare to uniform downsampling (referred to as "Uniform"), our deformed downsampling samples more densely at difficult regions (yellow arrows) and ignore image contents that are less important (green arrows). We also demonstrate our method on the street and aerial benchmarks (video and code link).

We begin this work with an experiment to investigate the limitations of the commonly employed uniform downsampling, and the current state-of-the-art technique, namely the "edge-based" downsampling (Marin et al., 2019). Motivated by the undesirable properties of these approaches, we then introduce *deformation module*, a learnable downsampling operation, which can be optimised together with the given segmentation model in an end-to-end fashion. Figure 2 provides an overview of the proposed method. The *deformation module* downsamples the high-resolution image over a non-uniform grid and generates a deformed low-resolution image, leading to its name. Moreover, to defend against degenerate solutions (e.g. over-sampling input images and labels at trivial regions like the backgrounds), we propose a regularisation term that encourages the sampling locations to concentrate around the object boundaries. We demonstrate the general utility of our approach on three datasets from different domains—Cityscapes (Cordts et al., 2016) and DeepGlobe (Demir et al., 2018)) and one medical imaging dataset—where we observe consistent improvements in the segmentation performance and the efficiency-and-accuracy trade-off.

## 2 METHODS

In this section, we first formulate the basic components associated with this study in Section 2.1. We then perform a motivation experiment in Section 2.2 to illustrate the impact of manual tuned objective of the "edge-based" downsampling approach (Marin et al., 2019), particularly the sampling density around edge, on segmentation performance and its spatial variation across the image. Motivated by the finding, we propose the *deformation module*, the *learnable downsampling module* for segmentation, in Section 2.3. To ease the joint optimisation challenge of oversampling at the trivial locations, a regularisation term is proposed in Section 2.4.

### 2.1 PROBLEM FORMULATION

**Sampling method.** Let $\mathbf{X} \in \mathbb{R}^{H \times W \times C}$ be a high-resolution image of an arbitrary size $H, W, C$. A sampler $G$ takes $\mathbf{X}$ as input and computes a downsampled image $\hat{\mathbf{X}} = G(\mathbf{X})$, where $\hat{\mathbf{X}} \in \mathbb{R}^{h \times w \times C}$.

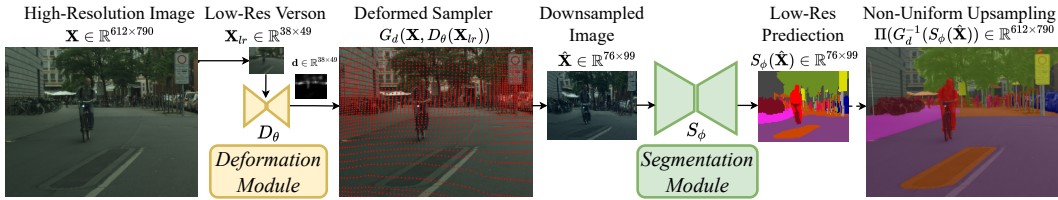

Figure 2: **Architecture schematic.** For each high-resolution image $\mathbf{X} \in \mathbb{R}^{H \times W \times C}$, we compute its lower resolution version $\mathbf{X}_{\text{lr}} \in \mathbb{R}^{h_d \times w_d \times C}$. The *deformation module*, $D_\theta$, parametrised by $\theta$, takes the low-resolution image $\mathbf{X}_{\text{lr}}$ as input and generates a deformation map $\mathbf{d} = D_\theta(\mathbf{X}_{\text{lr}})$, where $\mathbf{d} \in \mathbb{R}^{h_d \times w_d \times 1}$, that predicts the sampling density at each pixel location. Next, the deformed sampler $G_d$ is constructed by taking both $\mathbf{X}$ and $\mathbf{d}$ as input and computes the downsampled image $\hat{\mathbf{X}} = G_d(\mathbf{X}, \mathbf{d})$, where $\hat{\mathbf{X}} \in \mathbb{R}^{h \times w \times C}$ and sampling locations are shown as red dots masked on the image. The downsampled image $\hat{\mathbf{X}}$ is then fed into the segmentation network to estimate the corresponding segmentation probabilities $\hat{\mathbf{P}} = S_\phi(\hat{\mathbf{X}})$. During training the label $\mathbf{Y}$ is downsampled with the same deformed sampler to get $\hat{\mathbf{Y}} = G_d(\mathbf{Y}, \mathbf{d})$ and $\{\theta, \phi\}$ are jointly optimised by minimise the segmentation specific loss $\mathcal{L}_s(\theta, \phi; \mathbf{X}, \mathbf{Y}) = \mathcal{L}_s(\theta, \phi; \hat{\mathbf{P}}, \hat{\mathbf{Y}})$. At inference time, the low-resolution prediction $S_\phi(\hat{\mathbf{X}})$ is non-uniformly upsampled to the original space through deterministic reverse sampler $G_d^{-1}()$ and interpolation function $\Pi()$.

Consider a relative coordinate system[1] such that $\mathbf{X}[u, v]$ is the pixel value of $\mathbf{X}$ where $u, v \in [0, 1]$. And an absolute coordinate system such that $\hat{\mathbf{X}}[i, j]$ is the pixel value of $\hat{\mathbf{X}}$ at coordinates $(i, j)$ for $i \in \{1, 2, ... h\}, j \in \{1, 2, ... w\}$. Essentially, the sampler $G$ computes a mapping between $(i, j)$ and $(u, v)$. Practically, sampler $G$ contains two functions $\{g^0, g^1\}$ such that:

$$\hat{\mathbf{X}}[i, j] := \mathbf{X}[g^0(i, j), g^1(i, j)]^2 \tag{1}$$

**Segmentation and non-uniform upsampling.** In this work we discuss model agnostic downsampling and upsampling methods, therefore any existing segmentation model can be applied. Here we denote the segmentation network $S_\phi$, parameterised by $\phi$, that takes as an input a downsampled image $\hat{\mathbf{X}}$ and makes a prediction $\hat{\mathbf{P}} = S_\phi(\hat{\mathbf{X}})$ in the low-resolution space. During training label $\mathbf{Y}$ is downsampled by the same sampler $G$ to get $\hat{\mathbf{Y}} = G(\mathbf{Y})$ to optimise the segmentation network $S_\phi$ by minimising the segmentation specific loss $\mathcal{L}_s(\phi; \hat{\mathbf{P}}, \hat{\mathbf{Y}})$. At testing, the upsampling process consists of a reverse sampler $G^{-1}()$ that reverse mapping each pixel at coordinates $(i, j)$ from the sampled image $\hat{\mathbf{X}}$ back to coordinates $(u, v)$ in the high-resolution domain. Then an interpolation function $\Pi()$ is applied to calculate the missing pixels to get the final prediction $\mathbf{P} = \Pi(G^{-1}(S_\phi(\hat{\mathbf{X}})))$. The nearest neighbour interpolation is used as $\Pi()$ in this work.

**Disentangle the intrinsic upsampling error.** At test time, a typical evaluation of downsampled segmentation with *Intersection over Union* ($IoU$) is performed after non-uniform upsampling, between the final prediction $\mathbf{P}$ and the label $\mathbf{Y}$. Hence $IoU(\mathbf{P}, \mathbf{Y})$ incorporated both segmentation error and upsampling error. The upsampling error is caused by distorted downsampling, hence to improve the interpretability of the jointly learned downsampler, we propose to disentangle the intrinsic upsampling error from the segmentation error by assuming a perfect segmentation at the downsampled space and calculating the error introduced after upsampling. In specific we calculate $IoU(\mathbf{Y}', \mathbf{Y})$, where $\mathbf{Y}' = \Pi(G^{-1}(G(\mathbf{Y})))$, indicting the same non-uniform downsampling and upsampling processes applied to the label $\mathbf{Y}$.

## 2.2 MOTIVATIONAL STUDY: INVESTIGATING THE BARRIER OF THE SOTA

The "edge-based" approach (Marin et al., 2019), separately train a non-uniform down-sampler $G_e$ minimizing two competing energy terms: "sampling distance to semantic boundaries" and "sampling distance to uniform sampling locations", as the first and second term in equation 2, where $\mathbf{b}[i, j]$ is the spatial coordinates of the closest pixel on the semantic boundary. A temperature term $\lambda$ is used to balance the two energies, whose value is empirically recommended to 1 by Marin et al. (2019) and decided the sampling density around the edge.

---

[1]A relative instead of absolute coordinate system is selected for sampling be calculated in a continues space.

[2]The "uniform" approach will have sampler $G_u = \{g_u^0(i, j) = (i-1)/(h-1), g_u^1(i, j) = (j-1)/(w-1)\}$.

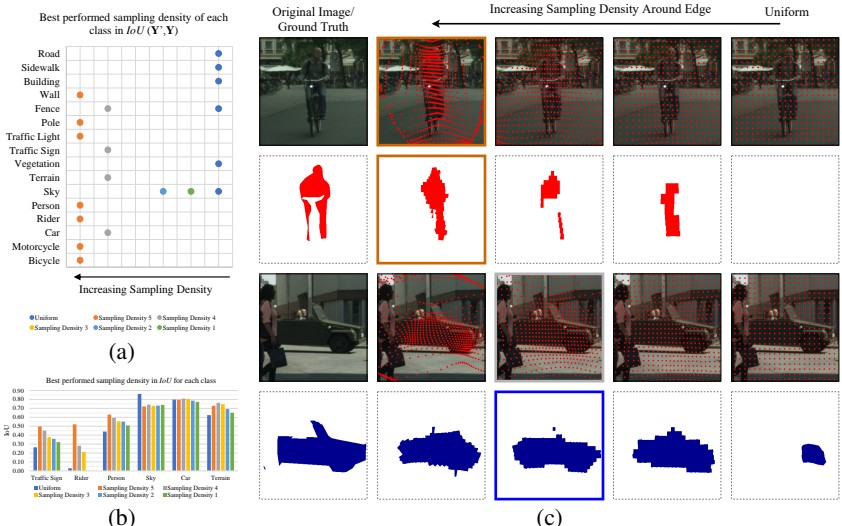

Figure 3: **Optimal sampling density varies over locations.** We simulating a set "edge-based" samplers each at different sampling density around edge, and evaluate class-wise performance in (a) $IoU(\mathbf{Y}',\mathbf{Y})$ and (b) $IoU$. Best performed sampling density varies for each class as in (a) and (b). Such variation also been observed visually in (c), where predictions with various sampling density for two example classes are illustrated. Sampling locations are masked in red dots and best performed prediction is highlighted. Motivational experiments are performed on binary segmentation on $1/10^{th}$ subset of Cityscapes (Cordts et al., 2016) (excludes under-represented classes) with downsampled size of $64 \times 128$ given input of $1024 \times 2048$.

$$E(G_e(i,j)) = \sum_{i,j} \|G_e(i,j) - \mathbf{b}[i,j]\|^2 + \lambda \sum_{|i-i'|+|j-j'|=1} \|G_e(i,j) - G_e(i',j')\|^2 \qquad (2)$$

The first part of our work performs an empirical analysis to investigate a key question: "How does the empirically tuned temperature term $\lambda$ affect the segmentation performance?". We hence perform binary segmentation on the Cityscapes dataset (Cordts et al., 2016). We generate a set of simulated "edge-based" samplers each at different sampling densities around the edge (i.e. $\lambda$) [3] and evaluate by either directly calculating $IoU(\mathbf{Y}',\mathbf{Y})$ given each fixed simulated sampler (Figure 3 (a)) or train a segmentation network with each fixed simulated sampler and evaluates its accuracy by $IoU$ (Figure 3 (b)). The results show there is no "one-size-fits-all" sampling density configuration that leads to the best performance for all individual classes, neither for intrinsic upsampling error ($IoU(\mathbf{Y}',\mathbf{Y})$ as in Figure 3 (a))) nor segmentation error ($IoU$ as in Figure 3 (b)), which is also verified visually in Figure 3 (c). These observations imply that the SOTA "edge-based" sampling scheme with a pre-set sampling density around the edge is sub-optimal, highlighting the potential benefits of a more intelligent strategy that can adapt sampling strategy to informative regions according to local patterns.

## 2.3 DEFORMATION MODULE: LEARN NON-UNIFORM DOWNSAMPLING FOR SEGMENTATION

Motivated by the above findings, we introduce the *deformation module*, a data-driven sampling method that adapts sampling density at each location according to its importance to the downstream segmentation task. Figure 2 provides a schematic of the proposed method.

Following earlier definitions in equation 1, to construct the deformed sampler $G_d$, we need to define the two sampling functions $\{g_d^0, g_d^1\}$ who inputs the downsampled coordinates $(i,j)$ in $\hat{\mathbf{X}}$ and the learned deformation map $\mathbf{d}$ to calculate the sampling location $(u,v)$ in $\mathbf{X}$, such that $\boldsymbol{u}[i,j] = g_d^0(i,j,\mathbf{d})$, $\boldsymbol{v}[i,j] = g_d^1(i,j,\mathbf{d})$. The idea is to construct the sampling functions to sample $\mathbf{X}$ denser at high-importance regions correspond to the salient regions in the deformation map $\mathbf{d}$. A naive approach is treating $\mathbf{d}$ as a sampling probability distribution and performing the stochastic sampling. However one has to estimate gradients through the reparameterisation trick and traverse stochastic

---

[3]The simulation details of the "edge-based" samplers is described in Appendix A.5.

sampling over all locations is not computationally efficient. Inspired by Recasens et al. (2018) we consider the sampling location of each low-resolution pixel $(i, j)$ is pulled by each surrounding pixel $(i^{'}, j^{'})$ by an attractive force, and the final sampling coordinates of $(i, j)$ can then be considered as a weighted average of the coordinates of each surrounding pixel $(i^{'}, j^{'})$. The weight at $(i^{'}, j^{'})$ is defined to be: 1) proportional to the sampling density $\mathbf{d}(i^{'}, j^{'})$; 2) degrade away from the centre pixel $(i, j)$ by a factor defined by a distance kernel $k((i,j),(i^{'},j^{'}))$ and 3) only applies within a certain distance. Practically the distance kernel is a fixed Gaussian, with a given standard deviation $\sigma$ and square shape of size $2\sigma + 1$, and the sampling coordinates of $(i,j)$ can then be calculated in the convolutional form by the two deformed sampler functions $\{g_d^0, g_d^1\}$ defined as equation 3. The standard deviation $\sigma$ decided the distance the attraction force can act on. While the magnitude of the attraction force is learned through the deformation map. We perform ablation studies to investigate the impact of $\sigma$ in Section A.1.

$$g_d^0(i,j,\mathbf{d}) = \frac{\sum_{i',j'} \mathbf{d}(i',j') k_\sigma((i,j),(i',j')) i'}{\sum_{i',j'} \mathbf{d}(i',j') k_\sigma((i,j),(i',j'))}, \quad g_d^1(i,j,\mathbf{d}) = \frac{\sum_{i',j'} \mathbf{d}(i',j') k_\sigma((i,j),(i',j')) j'}{\sum_{i',j'} \mathbf{d}(i',j') k_\sigma((i,j),(i',j'))} \quad (3)$$

This formulation holds certain desirable properties that fit our goal: 1) it computes sampling locations proportional to the learned deformation map that indicting varying sampling density at different regions, the need of which was illustrated in the motivational studies in Section 2.2; 2) the fixed Gaussian distance kernel can naturally fit CNN architecture, preserve differentiability and be efficient to calculate. This way the sampling density at each location is optimised by segmentation performance rather than separately trained based on manual designed objective as the "edge-based" approach (Marin et al., 2019)).

## 2.4 REGULARISATION OF THE JOINT-TRAINED SAMPLING

Our method is similar to Recasens et al. (2018), which however is only explored for image classification tasks. Transferring the method proposed in Recasens et al. (2018) to segmentation not only needs reformulating the problem as described in Section 2.1, optimising the joint system at the pixel level is not trivial. Given the segmentation loss $\mathcal{L}_s(\theta,\phi;\hat{\mathbf{P}},\hat{\mathbf{Y}}) = \mathcal{L}_s(\theta,\phi;\mathbf{X},\mathbf{Y})$ is calculated in the low-resolution space for minimal computation. The downsampling parameters $\theta$ can be optimised to trivial solutions which encourage oversampling both input image $\mathbf{X}$ and label $\mathbf{Y}$ at easy locations like the background to reduce loss, which contradicts our goal. We experimentally verified the naive adaptation of the method from Recasens et al. (2018) to segmentation (red bars in Figure 4) does not perform satisfactionally. To discourage the network learning trivial solutions, we propose a regularisation *edge loss* by encouraging the deformation map similar to a simulated target which leads to denser sampling around the object boundaries, inspired by Marin et al. (2019) that object edge may be informative for segmentation. Different to the approach of Marin et al. (2019) we don't manually tune optimal target deformation map but treat it as a regularisation term to carry its main message that edge information is useful, then let the joint training system adapts optimal sampling location according to the segmentation objective.

In specific, we calculate the target deformation map $\mathbf{d}_t = f_{edge}(f_{gaus}(\mathbf{Y}_{lr}))$[4] from the uniformly downsampled segmentation label $\mathbf{Y}_{lr} \in \mathbb{R}^{h_d \times w_d \times 1}$, which is at the same size as the deformation map $\mathbf{d}$. The regularisation *edge loss* is defined as $\mathcal{L}_e(\theta;\mathbf{X}_{lr},\mathbf{Y}_{lr}) = f_{MSE}(\mathbf{d},\mathbf{d}_t)$, which calculates the mean squared error (MSE) between the predicted deformation map $\mathbf{d} = D_\theta(\mathbf{X}_{lr})$ and the target deformation map $\mathbf{d}_t$. To this end, we jointly optimise the parameters $\{\theta,\phi\}$ corresponding to the *deformation module* and the *segmentation module* respectively by the single learning objective as equation 4, where the first term is the segmentation specific loss (e.g. cross entropy + $L2$ weight-decay) and the second term is the *edge loss*. We add a weight term $\gamma$ for both losses that have comparable magnitudes. We note different to $\lambda$ in equation 2, $\gamma$ can better adapt sampling location to difficult regions because it is balancing the edge-loss and segmentation loss (end task loss) in an end-to-end setting, while $\lambda$ is balancing the two manual designed sampling targets in a separately trained downsampling network. We also evaluate the impact of $\gamma$ in Section A.1.

$$\mathcal{E}(\theta,\phi;\mathbf{X},\mathbf{Y}) = \mathcal{L}_s(\theta,\phi;\hat{\mathbf{P}},\hat{\mathbf{Y}}) + \gamma \mathcal{L}_e(\theta;\mathbf{X}_{lr},\mathbf{Y}_{lr}) \quad (4)$$

---

[4]$f_{edge}(.)$ is an edge detection filter by a convolution of a specific $3 \times 3$ kernel [[-1, -1, -1], [-1, 8, -1], [-1, -1, -1]] and $f_{gaus}(.)$ is Gaussian blur with standard deviation $\delta = 1$ to encourage sampling close to the edge. To avoid the edge filter pickup the image boarder, we padding the border values prior to applying the edge filter.

## 3 RELATED WORKS

Convolutions with fixed kernel size restricted information sampled ("visible") for the network. Various multi-scale architectures have shown the importance of sampling multiple-scale information, either at the input patches (He et al., 2017; Jin et al., 2020) or feature level (Chen et al., 2014; 2016; Hariharan et al., 2015), but less efficient because of multiple re-scaling or inference steps. Dilated convolution (Yu & Koltun, 2015) shift sampling locations of the convolutional kernel at multiple distances to collect multiscale information therefore avoided inefficient re-scaling. Such shifting is however limited to fixed geometric structures of the kernel. Deformable convolutional networks (DCNs) (Dai et al., 2017) learn sampling offsets to augment each sampling location of the convolutional kernel. Although DCNs share similarities with our approach, they are complementary to ours because we focus on learning optimal image downsampling as pre-processing so that keeps computation to a minimum at segmentation and a flexible approach can be plug-in to existing segmentation networks.

Learned sampling methods have been developed for image classification, arguing better image-level prediction can be achieved by an improved sampling while keeping computation costs low. Spatial Transformer Networks (STN) (Jaderberg et al., 2015) introduce a layer that estimates a parametrized affine, projective and splines transformation from an input image to recover data distortions and thereby improve image classification accuracy. Recasens et al. (2018) proposed to jointly learn a saliency-based network and "zoom-in" to salient regions when downsampling an input image for classification. Talebi & Milanfar (2021) jointly optimise pixel value interpolated (i.e. super-resolve) at each fixed downsampling location for classification. However, an end-to-end trained downsampling network has not been proposed so far for per-pixel segmentation. Joint optimising the downsampling network for segmentation is more challenging than the per-image classification, as we experimentally verified, due to potential oversampling at trivial locations and hence we propose to simulate a sampling target to regularise the training.

On learning the sampling for image segmentation, post-processing approaches (Kirillov et al., 2020; Huynh et al., 2021) refining samplings at multi-stage segmentation outputs are complementary to ours, but we focus on optimising sampling at inputs so that keeps computation to a minimum. Non-uniform grid representation modifying the entire input images to alleviate the memory cost, such as meshes (Gkioxari et al., 2019), signed distance functions (Mescheder et al., 2019), and octrees (Tatarchenko et al., 2017). However none of those approaches is optimised specifically for segmentation. Recently Marin et al. (2019) proposed to separately train an "edge-based" downsampling network, to encourage denser sampling around object edges, hence improving segmentation performance at low computational cost and achieving state-of-the-art. However, the "edge-based" approach is sub-optimal with sampling density distribution fixed to manual designed objective, the distance to edge, rather than segmentation performance. We verify this point empirically in Section 2.2 and hence propose our jointly trained downsampling for segmentation.

## 4 EXPERIMENTS AND RESULTS

In this section, we evaluate the performance of our deformed downsampling approach on three datasets from different domains as summarised in Table 1, against three baselines. We evaluate the segmentation performance with $IoU$ and verify the quality of the sampler by looking at the intrinsic upsampling error with $IoU(\mathbf{Y}', \mathbf{Y})$. We show quantitative and qualitative results in Section 4.1. We further compare with reported results from Marin et al. (2019) in Section 4.2 with additionally experiments performed under the same preprocessing with the square central crop of input images.

**Model and training.** The *deformation module* is defined as a small CNN architecture comprised of 3 convolution layers. The segmentation network is defined as a deep CNN architecture, with HRNetV2-W48 (Sun et al., 2019) used in all datasets, and PSP-net (Zhao et al., 2017) as a sanity check in the Cityscapes dataset. We employ random initialisation like as in He et al. (2015), the same training scheme (Adam (Kingma & Ba, 2014), the focal loss (Lin et al., 2017) as the *segmentation loss* and MSE loss for the *edge-loss* with full details in Appendix A.7) unless otherwise stated.

**Baselines.** We compare two versions of our method either with single segmentation loss ("Ours-Single loss") or adding the *edge-loss* as equation 4 ("Ours-Joint loss") against three baselines: 1) the "uniform"

downsampling; 2) the "edge-based" [5] and 3) "interpolation", the jointly learned method for pixel value interpolation at fixed uniform downsampling locations (Talebi & Milanfar, 2021), as an additional performance reference in Section 4.2, whose implementation details are given in Section A.2.

Table 1: Dataset summary. More details in Appendix A.6

| Dataset | Content | Resolution (pixels) | Number of Classes |
|---|---|---|---|
| Cityscape (Cordts et al., 2016) | Urban scenes | $2048 \times 1024$ | 19 |
| DeepGlobe (Demir et al., 2018) | Aerial scenes | $2448 \times 2448$ | 6 |
| PCa-Histo (local) | Histopathological | $1968 \pm 216 \times 9392 \pm 4794$ | 6 |

### 4.1 QUANTITATIVE AND QUALITATIVE PERFORMANCE ANALYSIS

We plot $mIoU$ and $mIoU(\mathbf{Y}', \mathbf{Y})$ in Figure 4 comparing our method against "uniform" and "edge-based" baselines on all three datasets. Figure 5 further investigate $IoU$ at the per-class level, aiming to understand the impact of different object sizes reflected by different class frequencies. We look at the performance along with a set of different downsampling sizes in Figure 6, which is a key matrix indicating whether a downsampling strategy can enable a better trade-off between performance and computational cost.

In Figure 4, "Ours-Single loss", as a basic version of our proposed method with single segmentation loss, performs better than the "uniform" baseline on 2/3 tested datasets. Adding the proposed *edge loss*, "Ours-Joint loss" consistently performs best on all datasets with 3% to 10% higher absolute $mIoU$ over the "uniform" baseline. The performance improvement from "Ours-Single loss" to "Ours-Joint loss" represents the contribution of the *edge loss* within the proposed joint training framework. The variation of improvements over datasets suggests that the informativeness of edge is data-dependent, hence an end-to-end system is needed to better adapt sampling to content. Besides, the segmentation error ($mIoU(\mathbf{Y}',\mathbf{Y})$) does not always agree with the upsampling error ($mIoU$), indicting a separately trained sampler with manual designed sampling objective may fail.

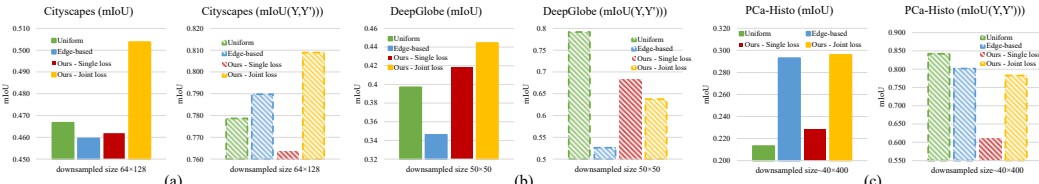

Figure 4: Comparing $mIoU$ and $mIoU(\mathbf{Y}', \mathbf{Y})$ of our joint trained downsampler, either with single segmentation loss ("Ours-Single loss") or additional edge loss ("Ours-Joint loss"), versus two baseline downsampling methods on three datasets.

At class-wise, as in Figure 5, "Ours-Joint loss" performs up to 25% better in absolute $IoU$ over baselines. Results also suggest the joint trained method can further generalise "edge-based" sampling for better segmentation performance. "Ours-Joint loss" not only improves the performances over all the low-frequency classes representing small objects, making use of the edge information as the "edge-based" baseline does, but also further generalise such improvements in the high-frequency classes representing large objects, where "edge-based" approach has been found difficult to improve over "uniform".

Good downsampling can enable a better trade-off between performance and computational cost. In Figure 6, our method shows can consistently save up to 90% of computational cost than the "uniform" baseline across a set of different downsampling sizes (details are given in Appendix Table 8) over all three datasets. We notice $mIoU$ does not increase monotonically on DeepGlobe dataset (Demir et al., 2018) with increased cost in Figure 6 (b), this indicates high resolution is not always preferable but a data-dependent optimal tradeoff exists and an end-to-end adjustable downsampler is needed. The $mIoU(\mathbf{Y}',\mathbf{Y})$ results show our method can adjust sampling priority depending on the

---

[5]Note this is an approximation of the method from Marin et al. (2019), which is the best-performed result from a set segmentation networks each trained with a simulated "edge-based" samplers represents using different $\lambda$, full details given in Appendix A.5.

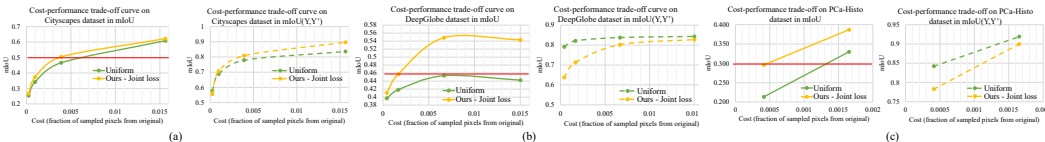

Figure 5: Class-wise IoU on three datasets. $IoU$ gain indicates improvements from our method over "uniform" baseline. Classes are ordered with increasing pixel frequency which is also indicated in each plot.

available computational resource, by compromising the upsampling error but focusing on improving segmentation at the low-cost end as in Figure 6 (b) and (c). Quantitative evidence all altogether shows our method can efficiently learn where to "invest" the limited budget of pixels at downsampling to achieve the highest overall return in segmentation accuracy.

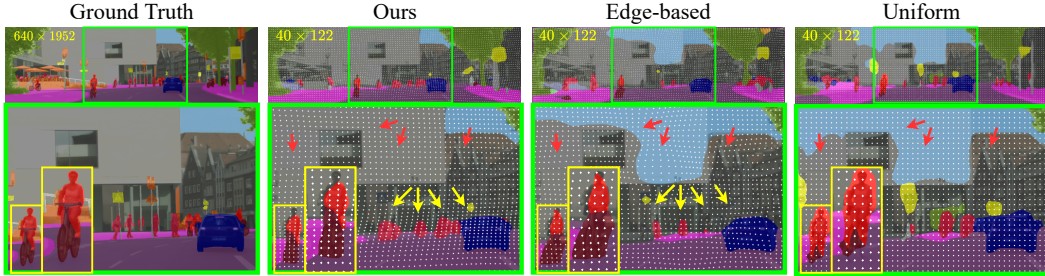

Figure 6: Cost-performance trade-offs in $mIoU$ and $mIoU(\mathbf{Y}',\mathbf{Y})$ on three datasets. Cost determines the amount of computation required at segmentation. The red lines indicates at same performance our method can save 33%, 90% and 70% from the "uniform" baseline on the three datasets respectively.

Visual results from Figure 7 and Figure 8 confirms our method integrates the advantage of the "edge-based" approach by sampling denser at small objects while further generalise with better sparse sampling on larger objects (see caption for details). Visual evidence also suggests the object edges are not always informative (see window edges in the wall class of Figure 7 misguided the edge-based sampler under-sampling of surrounding textures and lead to miss-prediction, and similarly as bushes in the green box of Figure 8). We therefore further discuss how boundary accuracy affects segmentation accuracy in the next section.

| Ground Truth | Ours | Edge-based | Uniform |
|---|---|---|---|

Figure 7: Examples on Cityscapes (Cordts et al., 2016) comparing our method against both baselines, where segmentation is performed on 16 times downsampled images (at each dimension). Predictions are masked over and sampling locations are shown in white dots. Yellow/ red arrows indicated regions denser/ sparser sampling helped to segment rider (red)/ sky (blue) classes, respectively.

## 4.2 A FAIR COMPARISON TO THE SOTA AND BOUNDARY ACCURACY ANALYSIS

In this section we perform experiments at same experimental condition on Cityscapes dataset and compare to reported results from Marin et al. (2019). The "interpolation" (Talebi & Milanfar, 2021) baseline is also compared. Figure 9 (a) shows our method outperforms all three baselines at all trade-off downsample sizes by upto 4% in absolute $mIoU$ over the second best result and achieves better cost-performance trade-off saving upto 47% calculation. Figure 9 (a) also suggests "interpolation" approach is less effective at small downsampling while its benefits increase as downsampling size increases. Visual comparison to "interpolation" baseline are provided in Appendix Section A.2.

We introduced the *edge-loss* to regularise the training of the joint system, inspired by Marin et al. (2019). Here we measure boundary precision and ask 1) how does it contribute to overall segmentation

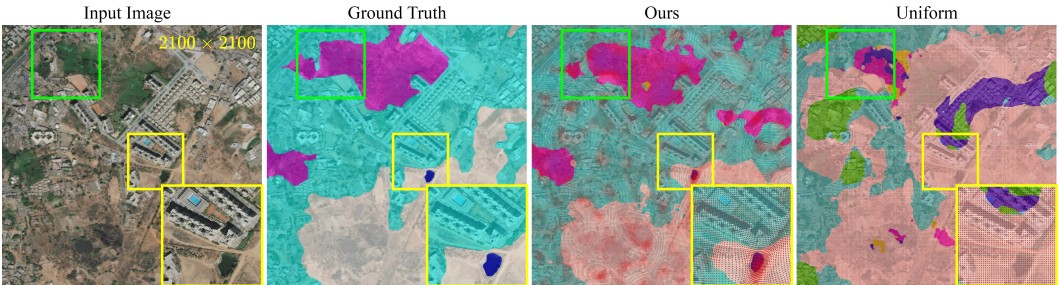

Figure 8: Qualitative example on DeepGlobe dataset (Demir et al., 2018) where segmentation performed on 8 times downsampled image (at each dimension). Predictions are masked over and sampling locations are shown in red dots. Yellow/ green boxed region indicated regions denser/ sparser sampling helped segmenting water (blue)/ forest (purple) classes, respectively.

accuracy? and 2) how differently do the joint trained system balance boundary accuracy and overall segmentation accuracy than the separately trained system (Marin et al., 2019)? We adopt trimap following Kohli et al. (2009); Marin et al. (2019) computing the accuracy within a band of varying width around boundaries in addition to $mIoU$, for all three datasets in Figure 9 (b). We found: 1) the "edge-based" baseline is optimal close to the boundary, but its performance does not consistently transfer to overall $mIoU$; 2) Our joint learned downsampling shows can identify the most beneficial sampling distance to invest sampling budget and lead to the best overall $mIoU$; 3) The most beneficial sampling distances learned by our approach are data-dependent that can close to the boundary (i.e. Cityscapes) or away (i.e. DeepGlobe and PCa-Histo); 4) it also suggests our method is particularly useful when the labelling is based on higher-level knowledge, i.e. Deepglobe and PCa-Histo dataset.

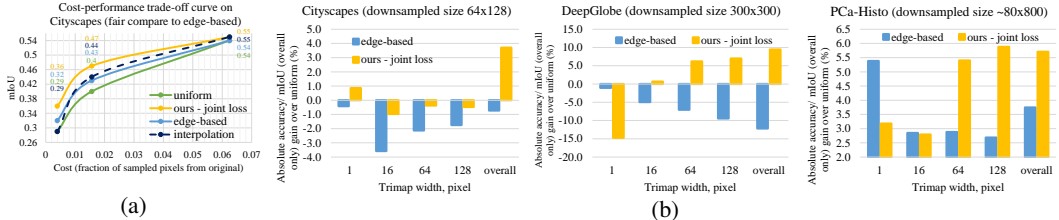

Figure 9: (a) Experiments performed at comparable condition to reported "edge-based" results by Marin et al. (2019) on Cityscapes dataset (Cordts et al., 2016), i.e. applying central $1024 \times 1024$ crop pre-processing and train with PSP-net (Zhao et al., 2017); (b) trimap analysis of absolute accuracy near semantic boundaries and overall $mIoU$ gain over "uniform" baseline on the three datasets.

## 5 CONCLUSION

We introduce an approach for learning to downsample ultra high-resolution images for segmentation tasks. The main motivation is to adapt the sampling budget to the difficulty of segmented pixels/regions hence achieving optimal cost-performance trade-off. We empirically illustrate the SOTA method (Marin et al., 2019) been limited by fixing downsampling locations to manual designed sampling objective, and hence motivate our end-to-end adaptive downsampling method. We illustrate simply extending the learn to downsample method from image classification (Recasens et al., 2018) to segmentation does not work, and propose to avoid learning trivial downsampling solutions by incorporating an edge-loss to regularise the training. Although our *edge-loss*, despite a simpler approximation, share the same spirit with Marin et al. (2019) we demonstrate our jointly trained system generalises sampling more robustly especially when object boundaries are less informative, hence consistently leading to a better cost-performance trade-off. Our method is light weighted and can be flexibly combined with the existing segmentation method without modifying the architecture.

ACKNOWLEDGMENTS

We would like to thank our reviewers for valuable discussions and feedback on the work and manuscript. We acknowledge the EPSRC grants EP/R006032/1, EP/M020533/1, the CRUK/EPSRC grant NS/A000069/1, and the NIHR UCLH Biomedical Research Centre which supported this research.

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

## A    APPENDIX

Here we provide additional results and various ablation studies and implementation details that have not been presented in the main paper.

## CONTENTS

### A.1    SENSITIVITY ANALYSIS ON HYPERPARAMETERS

We perform sensitivity analysis on four hyperparameters in our system, on Cityscapes (Cordts et al., 2016) and DeepGlobe (Demir et al., 2018) datasets in Table 2 to Table 5. In specific, Table 2 investigate the standard deviation $\sigma$ of the kernel $k$ in equation 3, which impact the distance the attraction force can be acted on. Table 3 and Table 4 investigates the impact of kernel size and input size on the proposed *deformation module*, which impact model capability and amount of context to the *deformation module*. Table 5 investigates edge-loss weight $\gamma$ in equation 4 which impact the weight of the *edge loss*. In general our method has been stable and robust across all tested hyperparameters (with absolute $mIoU$ variation within 3%).

Table 2: The sensitivity analysis of the hyperparameter of the kernel $k$ in equation 3: the standard deviation $\sigma$.

| Cityscapes ($1024^2$ to $96^2$ pixels) | | | |
|---|---|---|---|
| $\sigma$ (pixels) | 15 | 26 | 32 |
| $mIoU$ | 0.35 | 0.34 | 0.32 |
| DeepGlobe ($2448^2$ to $100^2$ pixels) | | | |
| $\sigma$ (pixels) | 20 | 25 | 33 |
| $mIoU$ | 0.43 | 0.45 | 0.43 |

Table 3: The sensitivity analysis of the hyperparameter of the kernel size in the proposed *deformation module*.

| Cityscapes ($1024^2$ to $96^2$ pixels) | | | |
|---|---|---|---|
| kernel size | $3{\times}3$ | $5{\times}5$ | $7{\times}7$ |
| $mIoU$ | 0.36 | 0.35 | 0.35 |
| DeepGlobe ($2448^2$ to $100^2$ pixels) | | | |
| kernel size | $3{\times}3$ | $5{\times}5$ | $7{\times}7$ |
| $mIoU$ | 0.45 | 0.45 | 0.46 |

Table 4: The sensitivity analysis of different low-res input sizes to the proposed *deformation module*.

| Cityscapes ($1024^2$ to $64^2$ pixels) | | | | | |
|---|---|---|---|---|---|
| low-res input size | $32^2$ | $48^2$ | $64^2$ | $80^2$ | $96^2$ |
| $mIoU$ | 0.35 | 0.35 | 0.34 | 0.33 | 0.36 |
| DeepGlobe ($2448^2$ to $50^2$ pixels) | | | | | |
| low-res input size | $50^2$ | $75^2$ | $100^2$ | $200^2$ | $300^2$ |
| $mIoU$ | 0.44 | 0.42 | 0.45 | 0.42 | 0.43 |

Table 5: The sensitivity analysis of the edge-loss weight $\gamma$ in equation 4

| Cityscapes ($1024^2$ to $64^2$) | | | |
|---|---|---|---|
| $\gamma$ | 50 | 100 | 200 |
| $mIoU$ | 0.33 | 0.35 | 0.35 |
| DeepGlobe ($2448^2$ to $100^2$) | | | |
| $\gamma$ | 50 | 100 | 200 |
| $mIoU$ | 0.45 | 0.45 | 0.43 |

A.2  IMPLEMENTATION AND VISUAL ILLUSTRATION OF THE "INTERPOLATION" BASELINE

The "interpolation" baseline is implemented by replacing our *deformation module* with the *resizing network* proposed by Talebi & Milanfar (2021) and without using the deformed sampler. The number of residual blocks ($r$) and the number of convolutional filters ($n$) are two key hyperparameters of *resizing network*, hence we perform each of our experiments with suggested hyperparameter combinations by Talebi & Milanfar (2021), and select the best performed results to represent the "interpolation" baseline, with all results provided in Table 6.

Table 6: The "interpolation" baseline performance with different combinations of the number of residual blocks ($r$) and the number of convolutional filters ($n$) for the *resizing network*, at each of the three downsampling sizes on the cityscapes dataset (with the central crop of size $1024 \times 1024$ pixels). Results are measured in $mIoU$.

| Downsampling size | Blocks / Filters | $r=1$ | $r=2$ | $r=3$ | $r=4$ |
|---|---|---|---|---|---|
| $64 \times 64$ | $n=16$ | 0.29 | 0.27 | 0.27 | 0.27 |
| | $n=32$ | 0.29 | 0.29 | 0.29 | 0.29 |
| $128 \times 128$ | $n=16$ | 0.42 | n/a | n/a | 0.43 |
| | $n=32$ | 0.43 | n/a | n/a | 0.44 |
| $256 \times 256$ | $n=16$ | 0.54 | n/a | n/a | 0.54 |
| | $n=32$ | 0.55 | n/a | n/a | 0.55 |

To better understand why our method works better than the "interpolation" baseline at small downsampling sizes as in Figure 9 (a), we plot visual examples in Figure 10. The visual examples confirm jointly learning where to sample is a more effective strategy when sampling budget is limited, while with more sampling budgets available the learning to "interpolation" approach would also work (Figure 9 (a)).

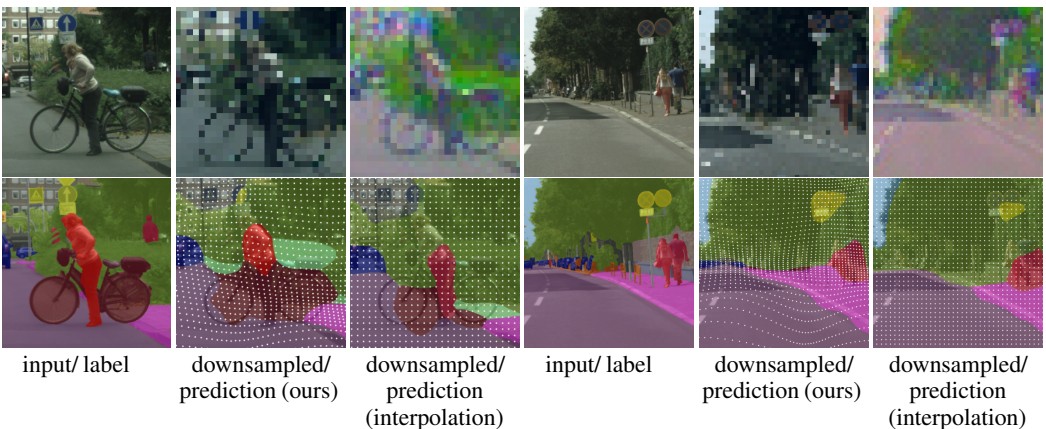

| input/ label | downsampled/ prediction (ours) | downsampled/ prediction (interpolation) | input/ label | downsampled/ prediction (ours) | downsampled/ prediction (interpolation) |

Figure 10: Visual comparison of our jointly trained downsampling against "interpolation" baseline (Talebi & Milanfar, 2021) on Cityscapes dataset where inputs of size $1024 \times 1024$ pixels are downsampled to $64 \times 64$ pixels. Sampling locations are masked over prediction as white dots.

A.3  MORE VISUAL EXAMPLES AT EXTREME DOWNSAMPLING RATES

A visual overview of applying our method on Cityscapes are given in Figure 11. To further verify how our method performs at extreme downsampling rate on histology images, in Figure 12, we show visual examples when a downsampling rate of 50 times at each dimension is applied to the PCa-Histo dataset. Consistent with quantitative results, our method can still perform well comparing to ground truth and significantly better than the "uniform" baseline at such an extreme downsampling rate. Our method performs especially well for the most clinically important classes, Gleason Grade 3 and Gleason Grade 4, the separation of these two classes is often referred to as the threshold from healthy to cancer. The sampling locations shown as blue dots in Figure 12 also indicate our method learned to sample densely at clinical important and challenge regions (e.g. cancerous nuclei and tissues) while sampling sparsely

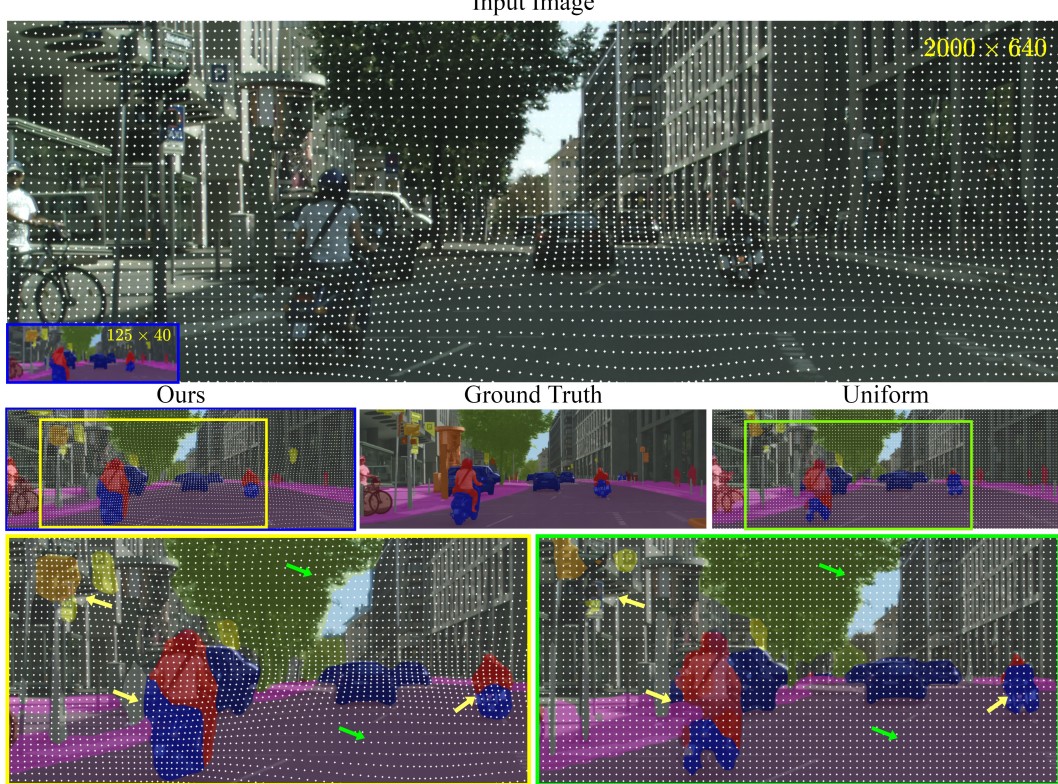

Figure 11: **Semantic segmentation with learned deformed downsampling on Cityscapes.** Our method learns non-uniform downsampling (white dots on the images) a high-resolution image (top-row) to preserve information that guides segmentation and ignore image content that does not contribute (green arrows) thereby producing a deformed downsampled image (bottom-left corner) which leads to more accurate low-resolution segmentation. This strategy helps identify small regions such as the people and traffic signs in the figure; see masked ground truth and segmentation reversed from low-resolution predictions in the middle-row. Compared to "Uniform" baseline our method ("Ours") sample more densely at important semantic regions (yellow arrows) and lead to a more accurate segmentation (see bottom-row).

at less informative regions (e.g. unlabelled/background), which lead to consistent improvement over uniform sampling in segmentation accuracy.

## A.4 INVESTIGATE THE LEARNED CLASS DISTRIBUTION

To verify whether our *deformation module* has learned to downsample denser at important regions/classes effectively, we monitor class frequency change after downsampling, as shown in Figure 13.

The results indicated the *deformation module* learned to adjust sampling strategy so that more pixels are "invested" to important regions/classes and less from the less informative regions/classes, and lead to optimal overall segmentation accuracy. For example, the *deformation module* learned to sample less (shown as negative sampling frequency ratio) in the "Background" class in the PCa-Histo dataset (Figure 13 (a)) and "road" and "sky" classes in the cityscapes (Cordts et al., 2016) dataset (Figure 13 (b)), in both cases those classes can be considered as potential "background" with common knowledge. However, our method also shows the ability to adjust sampling strategy on a more complex dataset where "background class" is less obvious. For example, in DeepGlobe (Demir et al., 2018) dataset there is no obvious "background class", but the "Urban" class has been learned to sample much denser than other classes (Figure 13 (c)), and leads to boosted segmentation accuracy (see Figure 4 and Figure 6 in the main paper). The learned distribution on the DeepGlobe dataset also verified our argument that the manual designed *sample denser around object edge strategy* as the "edge-based" (Marin et al., 2019) method may fail when object boundary is less informative, which is justified in the main results.

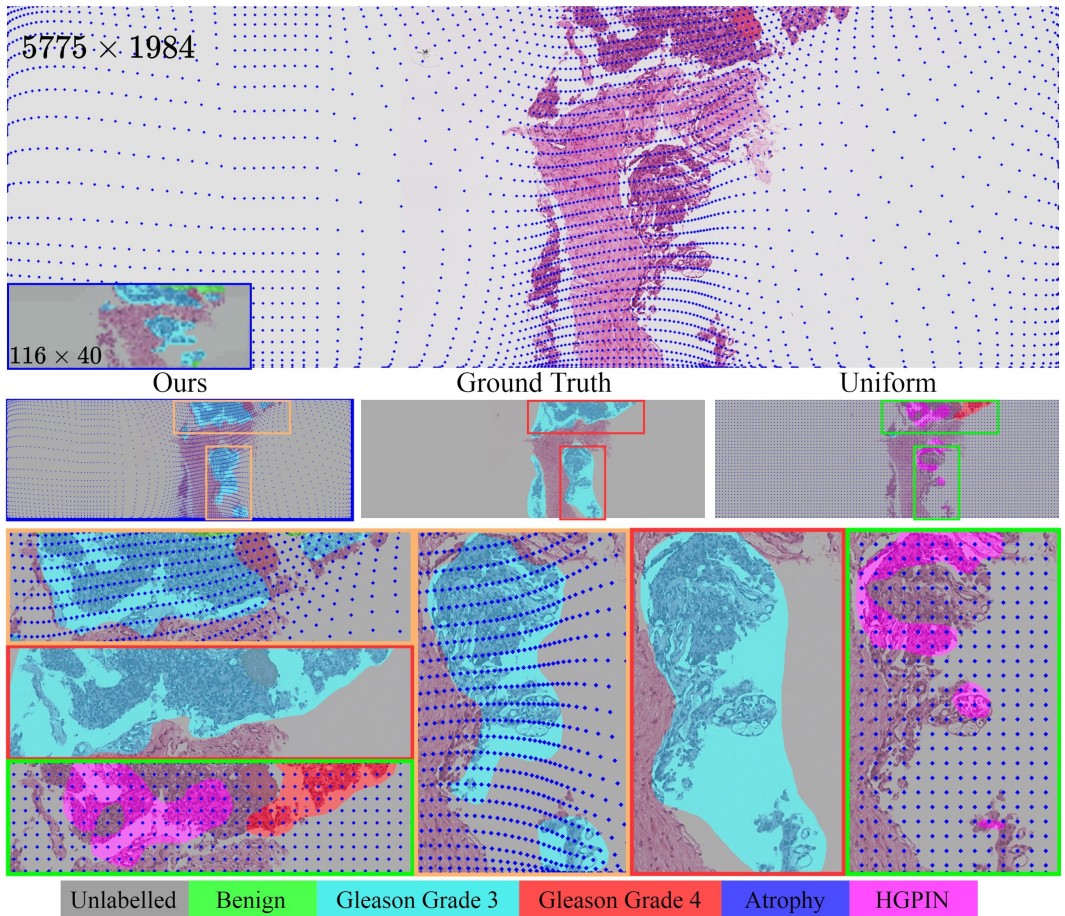

Figure 12: Qualitative example on PCa-Histo dataset (1) where segmentation performed on 50 times (at each dimension) downsampled image. Predictions are masked over and sampling locations are shown in blue dots.

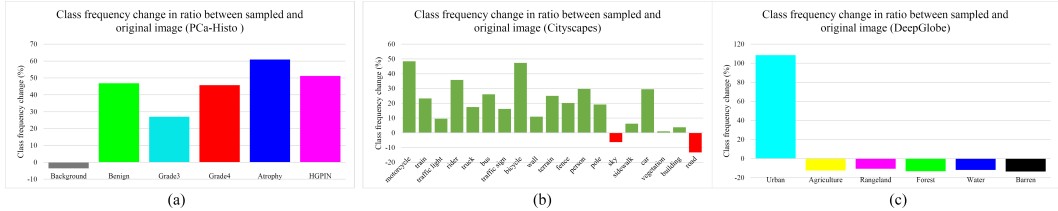

Figure 13: Class frequency change in the ratio between the sampled and original image. Higher value indicting more sampling pixels are "invested" in one class with the learned sampler. Results suggest our method can learn to sampler denser at semantic important classes while ignoring those not contribute. Results are from three datasets: (a) a local prostate cancer histology dataset (PCa-Histo) at the dynamic downsampled size of $80 \times 800$ pixels (b) Cityscapes (Cordts et al., 2016) at the downsampled size of $64 \times 128$ pixels and (c) DeepGlobe (Demir et al., 2018) at the downsampled size of $300 \times 300$ pixels.

## A.5 SIMULATED "EDGE-BASED" RESULTS

We direct simulate a set of "edge-based" samplers each with a different sampling density around edges. For each simulated "edge-based" downsampler, we expand sampling locations away from the label edge at different distances in a Gaussian manner to mimic the effect of $\lambda$ on sampling as shown by Marin et al. (2019). In specific, we calculate edge from label then apply gaussian blur to generate a simulated deformation map to guide sampling in our framework. We fix each simulated "edge-based"

downsamplers to train a segmentation network, with all results provided in Figure 14, among which best-performed in each dataset is selected as the "edge-based" baseline.

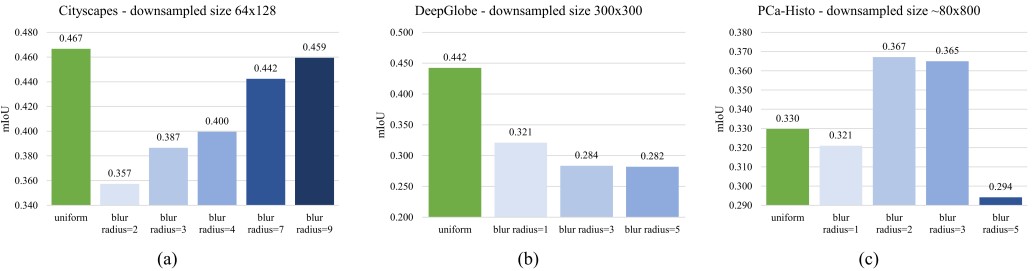

Figure 14: The segmentation accuracy with a set simulated "edge-based" downsamplers. Each bar is the segmentation accuracy of a segmentation network trained with one simulated "edge-based" downsampler. Each downsampler is simulated with a different sampling density around edges, represented by blur radius. The low blur radius indicates dense sampling around edges. The best-performed result is selected and referred to as the "edge-based" result presented in the main paper.

## A.6 DATASETS

In this work, we verified our method on three segmentation datasets: DeepGlobe aerial scenes segmentation dataset (Demir et al., 2018), Cityscapes urban scenes segmentation dataset (Cordts et al., 2016) and PCa-Histo medical histopathological segmentation dataset.

The **DeepGlobe (Demir et al., 2018)** dataset has 803 high-resolution ($2448 \times 2448$ pixels) images of aerial scenes. There are 7 classes of dense annotations, 6 classes among them are used for training and evaluation according to Demir et al. (2018). We randomly split the dataset into the train, validate and test with 455, 207, and 142 images respectively.

The **Cityscapes (Cordts et al., 2016)** dataset contains 5000 high-resolution ($2048 \times 1024$ pixels) urban scenes images collected across 27 European Cities. The finely-annotated images contain 30 classes, and 19 classes among them are used for training and evaluation according to Cordts et al. (2016). The 5000 images from the Cityscapes are divided into 2975/500/1525 images for training, validation and testing.

The **PCa-Histo:** dataset contains 266 ultra-high resolution whole slide images, from 33 distinct biopsies. The size of the images ranged from 250800 pixels to 40880000 pixels. Each pixel is annotated into one out of six classes: Unlabelled, Benign, Gleason Grade 3, Gleason Grade 4, Atrophy, HGPIN (see Table 7 for explanations of each class). Among the six classes, informative classes are underrepresented compared to Unlabelled classes (see Table 7). We random split the dataset into 200 training, 27 validation and 39 test images. This dataset has a varying size, with height (the short axis) range between 229 to 2000 pixels ($1967.5 \pm 215.5$ pixels), width (the long axis) range between 3386 to 20440 pixels ($9391.9 \pm 4793.8$ pixels) and Aspect Ratio (AP) range between 1.69 to 33.9 ($5.04 \pm 3.54$). The downsampling size of this dataset is calculated as follows: with a dynamic downsample size of for example $80 \times 800$, for each image, we calculate two candidate downsample sizes by either downsampling the short axis to 80 or the long axis to 800 while in either case we keep AP unchanged. Then we select the downsampling size that leads fewer total pixels. For example, for an image with a size $368 \times 7986$ pixels, we have candidate downsample sizes of $40 \times 868$ pixels or $36 \times 800$ pixels, and we will use downsample size of $36 \times 800$ in our experiment.

Table 7: Pixel frequency distribution of PCa-Histo dataset and explanation of each class

| Label | Background | Benign | Gleason Grade 3 | Gleason Grade 4 | Atrophy | HGPIN |
|---|---|---|---|---|---|---|
| % pixels in dataset | 93.99 | 0.97 | 1.23 | 3.35 | 0.26 | 0.20 |
| Explanation | all unlabelled pixels | healthy | mild cancer | aggressive cancer | healthy noise | precursor legion |

The prostate biopsies of the PCa-Histo dataset are collected from 122 patients. The identity of the patients is completely removed, and the dataset is used for research. The biopsy cores were sliced,

mounted on glass slides, and stained with H&E. From these, one slice of each of 220 cores was digitised using a Leica SCN400 slide scanner. The biopsies were digitised at 5, 10 and 20 magnification. At 20 magnification (which is the magnification used by histopathologists to perform Gleason grading), the pixel size is $0.55\,\mu m^2$.

## A.7 Network Architectures and Implementation Details

**Architectures:** The *deformation module* is defined as a small CNN architecture comprising 3 layers each with $3 \times 3$ kernels follower by BatchNorm and Relu. The number of kernels in each respective layer is $\{24, 24, 3\}$. A final $1 \times 1$ convolutional layer is applied to reduce the dimensionality and a softmax layer for normalisation are added at the end. All convolution layers are initialised following He initialization (He et al., 2015). The *segmentation module* was defined as a deep CNN architecture, with HRNetV2-W48 (Sun et al., 2019) applied in all datasets, and PSP-net (Zhao et al., 2017) as a sanity check in the Cityscapes dataset. The segmentation network HRNetV2-W48 is pre-trained on the Imagenet dataset as provided by Sun et al. (2019). In all settings, the size of the deformation map $\mathbf{d}$ is either same or twice the size of the downsampled image.

**Training:** For all experiments, we employ the same training scheme unless otherwise stated. We optimize parameters using Adam (Kingma & Ba, 2014) with an initial learning rate of $1 \times 10^{-3}$ and $\beta = 0.9$, and train for 200 epochs on DeepGlobe, 250 epochs on PCa-Histo dataset, and 125 epochs on Cityscapes dataset. We use a batch size of 4 for the Cityscapes dataset (Cordts et al., 2016) and the DeepGlobe dataset (Demir et al., 2018) and a batch size of 2 for the PCa-Histo dataset. We employ the step decay learning rate policy. We scale the *edge loss* 100 times so it is on the same scale as the *segmentation loss*. During the training of the segmentation network we do not include upsampling stage but instead, downsample the label map. For all datasets, we take the whole image to downsample as input, unless otherwise stated. During training, we augment data by random left-right flipping. All networks are trained on 2 GPUs from an internal cluster (machine models: gtx1080ti, titanxp, titanx, rtx2080ti, p100, v100, rtx6000), with syncBN.

**Downsampling sizes:** Downsampling configurations for the cost-performance trade-offs experiments are summarised in Table 8.

Table 8: Downsampling sizes of the cost-performance trade-offs experiments

| Cityscapes (original size $1024 \times 2048$ pixels) | | | | |
|---|---|---|---|---|
| downsampled sizes (pixels) | $16 \times 32$ | $32 \times 64$ | $64 \times 128$ | $128 \times 256$ |
| fraction of pixels sampled from original | $2e-4$ | $1e-3$ | $3.9e-3$ | $1.6e-2$ |
| DeepGlobe (original size $2448^2$ pixels) | | | | |
| downsampled sizes (pixels) | $50 \times 50$ | $100 \times 100$ | $200 \times 200$ | $300 \times 300$ |
| fraction of pixels sampled from original | $4e-4$ | $1.7e-3$ | $6.7e-3$ | $1.5e-2$ |
| PCa-Histo (original size $1968 \pm 216 \times 9392 \pm 4794$ pixels) | | | | |
| downsampled sizes (pixels) | $40 \times 400$ | | $80 \times 800$ | |
| fraction of pixels sampled from original | $4.1e-4$ | | $1.7e-3$ | |

