# OpenReview forum: "Learning to Downsample for Segmentation of Ultra-High Resolution Images"
_ICLR.cc/2022/Conference — ICLR 2022 Poster_

### Official Review · Reviewer_eqeg · 2021-10-30

**Correctness:** 4
**Technical Novelty And Significance:** 3
**Empirical Novelty And Significance:** 2
**Recommendation:** 6
**Confidence:** 3

**Main Review:**

Strengths:

1. The paper is well-written. The problem, motivation, idea, and method are clearly stated. Sufficient visualization is provided to make the paper easier to read.

2. Comprehensive experiments and analyses are conducted. The results on three publicly available datasets are reported.



Weaknesses:

1. As the authors said, the proposed method is similar to [Recasens et al. 2018], so the novelty of the proposed method is somewhat limited. However, the authors show that directly applying [Recasens et al. 2018] to segmentation tasks is not necessarily effective. Hence, I agree with the authors that the proposed method is somewhat novel.


**Summary Of The Paper:**

This paper address the semantic segmentation problem on high-resolution images. Existing methods uniformly downsample the original image to a small version to meet the memory requirement, while the uniform downsampling is suboptimal. The authors propose a deformed downsampling method in this paper and mainly compare it with the previous edge-based downsampling, showing better performance. Although the proposed method is somewhat similar to one previous method, while directly applying which on segmentation task does not improve the performance. This demonstrates the importance of the modifications proposed by the authors in this paper.

**Summary Of The Review:**

The paper is well-argued, and the proposed method is effective. Novelty is somewhat limited but can meet the ICLR bar.

---

> ### Author Response · Authors · 2021-11-20
> **Respond to R4 (eqeg)**
>
> We thank the reviewer for the thoughtful reviews!
>
> We appreciate the reviewer’s concern on technical novelty for the method is similar to [Recasens et al. 2018]. Despite adapting [Recasens et al. 2018] from image-level classification to pixel-level segmentation is challenging that the reviewer agreed. We also highlight the following contributions of this work:
>
> - Our work is the first end-to-end sampling scheme that adapts to a memory budget according to the difficulty of the pixel-level tasks, which bring new ideas to many relevant vision tasks as pointed out by all other three reviewers;
> - Following all reviewer’s constructive feedback, we added one new baseline model (Talebi, H., and Milanfar, P., ICCV 2021) and four more sensitivity analyses on hyperparameters and further justified our method is solid;

---

### Official Review · Reviewer_EMME · 2021-11-02

**Correctness:** 3
**Technical Novelty And Significance:** 3
**Empirical Novelty And Significance:** 3
**Recommendation:** 6
**Confidence:** 2

**Main Review:**

The strengths of the papers are:
1. It extends the learn to zoom approach for classification to segmentation tasks. A deformable module is applied for downsampling the original image.
2. The upsampling processing is incorporated in the training phase.
3. The edge information is also used to guild the downsampling step which improves the segmentation performance.

The overall presentation of the method is great. I have some questions about the paper.
1. The loss function consists of segmentation loss and edge loss. A weight term is used to balance these two terms. Is this the same problem as mentioned in Marin et. al (2009) in the barrier of the SOTA section?
2. It would be better to add more text to generate the downsampled image (ex. R^{52x100}) from low-res (ex. R^{13x25}).

Missing references:
Kirillov, Alexander et. al., Pointrend: Image segmentation as rendering, CVPR 2020
Huynh, Chuong et. al., Progressive Semantic Segmentation, CVPR 2021

**Summary Of The Paper:**

The authors present a method for ultra-high resolution image segmentation. It follows the learn to zoom approach to non-uniform downsample the original high resolution image. Furthermore, it adds edge-based loss to guide the downsample process. The upsampling step is also included in the training stage. The proposed method is evaluated on several public datasets and achieves promising results. Effectiveness of components are verified in the experiments.

**Summary Of The Review:**

The method mainly extends the learn to zoom approach for classification to segmentation tasks. Importantly, the edge loss is applied to guild the learning based downsample. The authors present extensive experiments and analysis to assess the components of the method. This method brings new ideas or approaches to handle high resolution images, especially in segmentation tasks.

---

> ### Author Response · Authors · 2021-11-20
> **Respond to R3 (EMME)**
>
> We thank the reviewer’s constructive feedback and answer specific questions below:
>
> > *The loss function consists of segmentation loss and edge loss. A weight term is used to balance these two terms. Is this the same problem as mentioned in Marin et. al (2009) in the barrier of the SOTA section?*
>
> - We clarify the difference in the revised methods section 3.4, particularly Eq.4: our weight is **balancing the edge-loss and segmentation loss** in the proposed **end-to-end** system so sampling is still **adapted to the end segmentation task**, while in Marin et,al. (2009) the weight is **balancing the two manual designed sampling targets** (close to edge versus close to uniform sampling locations) in a **separately trained** downsampling network hence sampling is **independent of the end segmentation task**. Therefore our sampling is less dependent on the weight and can better adapt sampling location according to the difficulty of the downstream pixels-level task;
> - Despite the above difference, we agree it’s an important term and hence **added sensitivity analysis** comparing with 0.5x and 2x of original weights we have applied in Table 5 in the revised Appendix and summarise results below in Table R3 - 1. We found **performance are stable** with variation within 2% absolute mIoU which justified our system is robust and less dependent on the weight $\gamma$.
>
> **Table R3 - 1**
>
> |  Cityscapes (1024x1024 downsample to 64x64) |      |      |      |
> |:-------------------------------------------:|:----:|:----:|:----:|
> |                   $\gamma$                  | 0.5x |  1x  |  2x  |
> |                     mIoU                    | 0.33 | 0.35 |      |
> | DeepGlobe (2448x2448 downsample to 100x100) |      |      |      |
> |                   $\gamma$                  | 0.5x |  1x  |  2x  |
> |                     mIoU                    | 0.45 | 0.45 | 0.43 |
>
> > *It would be better to add more text to generate the downsampled image (ex. R^{52x100}) from low-res (ex. R^{13x25}).*
> - We agree and **added sensitivity analysis** on different low-res input sizes in Table 4 in the revised Appendix, and summarise results below in Table R3 - 3. We tested a set of low-res input sizes between 0.5x to 6x of downsampled image size on two datasets and found 1) **performance been stable** with variation within 3% absolute mIoU which justified our system is robust to low-res input size; 2) **larger low-res input size not necessarily improve performance** demonstrating overly providing information to the downsampling network can add noise and reduce performance.
>
> Table R3 - 2
>
> | Cityscapes (1024x1024 downsample to 64x64) |      |      |      |      |      |
> |:------------------------------------------:|:----:|:----:|:----:|:----:|:----:|
> |     low-res input size (pixels square)     |  32  |  48  |  64  |  80  |  96  |
> |                    mIoU                    | 0.35 | 0.35 | 0.34 | 0.33 | 0.36 |
> |  DeepGlobe (2448x2448 downsample to 50x50) |      |      |      |      |      |
> |     low-res input size (pixels square)     |  50  |  75  |  100 |  200 |  300 |
> |                    mIoU                    | 0.44 | 0.42 | 0.45 | 0.42 | 0.43 |
>
> > *Missing references: Kirillov, Alexander et. al., Pointrend: Image segmentation as rendering, CVPR 2020 Huynh, Chuong et. al., Progressive Semantic Segmentation, CVPR 2021*
> - We thank the reviewer for pointing out these very relevant references, which we added in the revised related work section 2.3. That “post-processing approaches (Kirillov et al., 2020; Huynh et al., 2021) refining samplings at multi-stage segmentation outputs are complementary to ours, but we focusing on optimising sampling at inputs so that keeps computation to a minimum.”

---

### Official Review · Reviewer_wmVs · 2021-11-03

**Correctness:** 3
**Technical Novelty And Significance:** 2
**Empirical Novelty And Significance:** 2
**Recommendation:** 6
**Confidence:** 3

**Main Review:**

Strength:
- This paper proposes a new deformed sampling module that explicitly downscale the image into target resolution. The idea of making the network to learn the sampling density is very intuitive, and it is also adaptable to classical sampling functions.
- Besides, the author noticed the importance of regularization and added the edge loss in training.
- The proposed method is end-to-end trainable, and can be plugged into different backbone networks.

Weakness:

Still, I have complex feeling about this paper: the motivation is really good: following all the previous works that try to improve the segmentation performance by allocating the computations to most important regions, the proposed module does make a lot of sense, and it's plug-and-play. But when I read the experiment results, I find that it is not the module itself that brings the most difference, but the joint loss. From Fig.5, it can be seen that adding the module trained with single loss is not as good as the uniform one at the most time. This implies that the most important contribution of this paper is the joint loss.

Though, if this paper wants to illustrate the importance of joint loss that avoids the trivial solution, there are still some missing points to support this claim:
- Not enough ablation studies. If applying the joint loss on other methods, how the performance would be influenced?
- More analysis of why the single segmentation loss would not work. After reading this paper, I still do not understand: why single loss would lead to a trivial solution?

I read the appendix and find the downsampling module consists of 3 layers of CNN with 3x3 kernel. Since the other works in 2.1 all aim to cover multiple scale information, does it matter that this module only has this limited spatial window? And, Fig. 12 shows that the sampling density suddenly increases on the edge of the picture, even if the content is still solid-color. Why would this happen? Is this learned module vulnerable to noises in image?

Besides, since this work adopts a learned CNN module to predict the sampling density, and then conduct rescaling by interpolation, I wonder why not directly makes the network to directly downsample/super-resolve the input? Such ideas are applied in "Learning to Resize Images for Computer Vision Tasks". How does the proposed method compare with this paper?


Minor errors:
- In the caption of Fig. 5, the last line should note "(c) a local prostate cancer histology dataset...".

**Summary Of The Paper:**

This paper aims to improve the performance of segmenting ultra high-resolution images by replacing the commonly used downsampling functions with the learned deformation sampling module. The biggest difference of the proposed method is, instead of sampling all areas uniformly, the sampling density is estimated based on the contents. The experiment results on the Cityscape, DeepGlobe and PCa-Histo datasets show some improvements on mIoU and cost-performance trade-offs compared with uniform downsampling.

**Summary Of The Review:**

Overall, this paper has good motivation and proposes an interesting method. But after reading the whole paper, I felt that it is not the proposed method, but the joint loss that contributes most to the performance improvement. Besides, this paper lacks of in-depth explanation of the performance drop when applying the single loss, and comparisons of several similar methods. So I would give score 5 for now.

---

> ### Author Response · Authors · 2021-11-20
> **Respond to R2 (wmVs) - Part 2**
>
> **Lack of ablation and comparing similar method:**
> > *“comparisons of several similar methods”*
>
> > *“Besides, since this work adopts a learned CNN module to predict the sampling density, and then conduct rescaling by interpolation, I wonder why not directly makes the network to directly downsample/super-resolve the input? Such ideas are applied in "Learning to Resize Images for Computer Vision Tasks". How does the proposed method compare with this paper?”*
>
> - We thank the reviewer for pointing out this very relevant reference,and clarify its theoretical difference to our work: **1) Targeting on different tasks** --- image-level classification in theirs versus pixel-level segmentation in ours; **2) Different optimisation goals**: they jointly optimise pixel value interpolated (i.e. super-resolve) at each fixed downsampling location, while we jointly optimise downsampling locations rather than the pixel value; **3)** As all reviewers acknowledged, learning to sample denser at difficult regions is **intuitive for pixel-level segmentation tasks**.
> - We also **implemented this method and added it as the third baseline**, referred to as “interpolation”, in revised Figure 10 (a) and visualisations in Section A.2 in the revised Appendix, and we summarise the results in the following Table R2 - 1. We observe 1) ours is still the best over all three baselines at the three different downsampling sizes; 2) the interpolation method is less effective at small downsampled size (64x64) while its benefits increase as downsampling size increases. The results justified our downsampling strategy is more effective at limited memory budget, hence leading to a better cost-performance tradeoff.
>
> **Table R2 - 1**: Cost-performance trade-off curve on Cityscapes in mIoU (original resolution 1024x1024)
>
> | downsampled size  | uniform | edge-based | interpolation | ours |
> |:---:|:---:|:---:|:---:|:---:|
> | 64x64 | 0.29 | 0.32 | 0.29 | **0.36** |
> | 128x128 | 0.40 | 0.43 | 0.44 | **0.47** |
> | 256x256 | 0.54 | 0.54 | **0.55** | **0.55** |
>
> - Visual comparison to the "interpolation" approach is also provided in Appendix Section A.2, which intuitively confirmed when the sampling **budget is limited, jointly learning where to “invest”  the limited sampling locations** is a more effective strategy, while with more sampling budgets available learn interpolation would also work.
> - We added the above clarifications in the revised related works, results and Appendix.
>
> **Other questions:**
> > *“I read the appendix and find the downsampling module consists of 3 layers of CNN with 3x3 kernel. Since the other works in 2.1 all aim to cover multiple-scale information, does it matter that this module only has this limited spatial window?”*
>
> - The **inputs** to the downsampling module are **low-resolution versions** of original input (uniformly downsampled) at small sizes (e.g. 64x128 for cityscapes), and we **empirically find** that a downsampling module with **3 layers of 3x3 CNN is sufficient** to capture the context variation;
> - However, we agree it is worth further checking how multiple-scale information affects the results, so we **add a sensitivity analysis** with 5x5 and 7x7 kernels to compare with the proposed 3x3 kernels on two datasets in Table 3 of the revised Appendix, and the results are copied in below Table R2-2. We found 1) **performance are stable** with variation within 1% absolute mIoU and 2) performance does **not necessarily increase with kernel size**. These results justified our system is robust and the proposed 3x3 kernel is sufficient.
>
> **Table R2-2**
>
> |  Cityscapes (1024x1024 downsample to 96x96) |      |      |      |
> |:-------------------------------------------:|:----:|:----:|:----:|
> |                 kernel size                 |  3x3 |  5x5 |  7x7 |
> |                     mIoU                    | 0.36 | 0.35 | 0.35 |
> | DeepGlobe (2448x2448 downsample to 100x100) |      |      |      |
> |                 kernel size                 |  3x3 |  5x5 |  7x7 |
> |                     mIoU                    | 0.45 | 0.45 | 0.46 |
>
>
> > *And, Fig. 12 shows that the sampling density suddenly increases on the edge of the picture, even if the content is still solid-color. Why would this happen? Is this learned module vulnerable to noises in images?*
> - This is an undesirable bias when simulating the target deformation map, where the **edge filters mistakenly pick the image border** as object edges. It **can be avoided by padding the border values** prior to applying the edge filter. We add clarification on this in revised methods section 3.4 where the target deformation map is defined.
>
> > *In the caption of Fig. 5, the last line should note "(c) a local prostate cancer histology dataset...".*
> - We thank the reviewer for pointing this out and corrected in the revised version.

---

> > ### Comment · Reviewer_wmVs · 2021-11-22
> > **Thanks for the detailed answers.**
> >
> > I have read the author's comment to all reviewers and the answer to my questions. My concerns are clearly addressed in the above answers, and the author even includes more ablation studies to examine the module design. Thus, I will change my recommendation to 6.

---

> > > ### Author Response · Authors · 2021-11-22
> > > **Thanks!**
> > >
> > > Thank you and the other reviewers for the extremely helpful suggestions!

---

> ### Author Response · Authors · 2021-11-20
> **Respond to R2 (wmVs) - Part 1**
>
> We thank the reviewer for the constructive feedback. However, we feel there is some misunderstanding, which we address while answering each question below:
>
> **Explain why joint training with single segmentation loss does not work?**
> > *“Besides, this paper lacks an in-depth explanation of the performance drop when applying the single loss”*
>
> > *“More analysis of why the single segmentation loss would not work. After reading this paper, I still do not understand: why single loss would lead to a trivial solution?”*
>
> - The single loss does not work well because the jointly trained downsampler is applied to **both input image and label** during training. This allows the segmentation loss to be calculated in low-resolution space, and reverse upsampling to occur only at inference so that keeps **computation to a minimum**, which is consistent with previous work (Marin et,al., 2019). However, this would **encourage oversampling both image and label** at easy pixels/ regions to minimise the single segmentation loss, which contradicts the intention to sample densely at difficult regions.
> - Such observations led us to propose the second **edge-loss at the output of the downsampling network** to regularise training by encouraging more sampling around the object edges. We revise the methods section (especially section 3.4) to clarify this.
>
> **Confusion on the “Ours - Joint loss” results.**
> > *“But after reading the whole paper, I felt that it is not the proposed method, but the joint loss that contributes most to the performance improvement.”*
>
> > *“But when I read the experiment results, I find that it is not the module itself that brings the most difference, but the joint loss.*
>
> - We clarify the “Ours - Joint loss” results represent the compound contribution from both the **proposed joint training system** and the **“joint loss” (i.e. segmentation loss + edge loss)**, which are the two key contributions of this work, and **neither of them alone would work**;
> - The **“joint loss” is not separable from the joint training system**, but we made our best attempts to estimate their independent contribution in the original submission, see the answers in the next question;
>
> > *From Fig.5, it can be seen that adding the module trained with single loss is not as good as the uniform one at the most time. This implies that the most important contribution of this paper is the joint loss.”*
>
> - In Fig.5, “Ours - Single loss” represents the independent contribution of the proposed joint training system with single segmentation loss and it does **perform better** than “uniform” baseline in mIoU on both DeepGlobe and PCa-Histo (i.e. **2/3 tested datasets**);
> - The contribution of the “**joint loss**” (i.e. segmentation loss + edge loss) **cannot be independently evaluated** in other non-joint-trainable-system because the **edge loss** only backpropagates the **downsampling module** and cannot be evaluated with only the segmentation module;
> - While the “joint loss” is not separable, the contribution of the edge-loss in a separately trained downsampling network can be evaluated, as the “edge-based” results in Fig.5 demonstrates, that the **edge-loss alone without a joint system cannot work flexibly**;
> - Therefore it’s the **right combination** of our two key contributions, the **joint training system** and the “**joint loss**”, that is helpful. we revised the experiments and results section, particular section 4.1 to clarify the above points.
>
> > *“If applying the joint loss on other methods, how the performance would be influenced?”*
>
> - The previous downsampling for segmentation methods is not compatible with the “joint loss”. The “joint loss” (i.e. segmentation loss + edge loss) is built on **our proposed jointly trained downsampling for segmentation method**, which is **the first jointly trained downsampler for the pixel-level task** to the best of our knowledge, and cannot be applied separately trained segmentation networks nor downsampling networks;

---

### Official Review · Reviewer_wnXt · 2021-11-04

**Correctness:** 4
**Technical Novelty And Significance:** 3
**Empirical Novelty And Significance:** 3
**Recommendation:** 8
**Confidence:** 3

**Main Review:**

+ The paper is well motivated based on detailed description of the limitations of the closest recent work.
+ The results are detailed and well explained.

- "Consider a relative coordinate system such that I[u, v] is the pixel value of I where u, v∈[0,1]." is not clear that why the coordinates need to be normalized between [0,1]
- The term b[i,j] in Eq.2 is not explained. Although its an equation from another paper, it would help to have some definition of this variable.
- Eq 3 needs to be explained a bit better. Also what is the kernel k in Eq.3 and how is it defined in the paper implementation.

**Summary Of The Paper:**

Semantic segmentation using deep learning techniques becomes challenging if the input images are of high resolution. The image resolution can be lowered and the network learned, but downsampling based on uniform grid pattern on the high resolution image can lead to network not being able to learn semantic information in high frequency regions, i.e. the loss of information is uniform over the image. In order to alleviate this problem, the downsampling can be done based on a non-uniform grid on the high resolution image which is content adaptive where more samples are selected at the high frequency/important regions while low frequency regions are sampled sparsely. Previous work has computed this non-uniform downsampling in a deep learning framework as an independent task. This paper proposes to jointly learn the downsampling grid estimation task and the particular task of semantic segmentation. They show improved IoU metric over semantic segmentation results on standard benchmarks.

**Summary Of The Review:**

The paper shows that jointly doing content driven subsampling and the high level task (semantic segmentation here) can lead to better results rather than doing them independently. This idea may be propagated to other high level vision tasks which are limited by image size of training data. Thus, the reviewer is inclined favorably towards this paper.

---

> ### Author Response · Authors · 2021-11-20
> **Respond to R1 (wnXt)**
>
> We are happy to see that the reviewer agreed on the challenge of segmenting high-resolution images and finds our work well-motivated with detailed results, and particularly:
>
> - Confirmed our empirical findings that using either **uniform** or **independently** trained downsampling is **not sufficiently** flexible for pixel-level segmentation tasks;
> - Agreed that our proposed approach of **jointly learning content-driven downsampling** with segmentation offers a significant advantage;
> - Pointing out the idea has potential **broader impact** through propagation to **other high-level vision tasks** which are limited by image size;
>
> We are also grateful for the reviewer pointing out certain issues of clarity which we revise and clarify as follows:
>
> > *"Consider a relative coordinate system such that I[u, v] is the pixel value of I where u, v∈[0,1]." is not clear that why the coordinates need to be normalized between [0,1]*
>
> - The purpose is to create a continuous coordinate system rather than discrete, to be consistent with the sampling location calculation in Eq.3. We clarify this point in the revised methods section 3.1.
>
> > *The term b[i,j] in Eq.2 is not explained. Although its an equation from another paper, it would help to have some definition of this variable.*
>
> - b[i,j] is the spatial coordinates of the closest pixel on the semantic boundary. We clarify this in the revised methods section 3.2 when introducing Eq.2.
>
> > *Eq 3 needs to be explained a bit better. Also, what is the kernel k in Eq.3 and how is it defined in the paper implementation.*
>
> - We revised the methods section 3.3 for clarification of Eq.3 and kernel k. The following from revised methods section 3.3 (3rd paragraph) answered specifically this question: “The idea is to construct the deformed sampler $G_{d}$ who samples $\mathbf{I}$ denser at high-importance regions based on the sampling density predicted in the deformation map $\mathbf{d}$. Inspired by Recasens et al. (2018) we consider the sampling location of each pixel ($\textit{i, j}$) is pulled by each surrounding pixel ($i^{'}, j^{'}$) by an attractive force. The attractive force is defined to be: 1) proportional to the sampling density $\mathbf{d}(i^{'}, j^{'})$; 2) degrade away from the center pixel ($\textit{i, j}$) by a factor defined by a distance kernel $k((i, j), (i^{'}, j^{'}))$ and 3) applies within a certain distance. Practically the distance kernel is a fixed Gaussian, with a given standard deviation $\sigma$ and square shape of size $2\sigma+1$. The composition of forces applied to a central pixel ($i, j$) can then be calculated in the convolutional form hence the two deformed sampler functions $\{g_{d}^{0}, g_{d}^{1}\}$ are defined as equation 3. The standard deviation $\sigma$ decided the distance the attraction force can act on, and the degree of force is learnt through the deformation map.”
>
> - We also added a sensitivity analysis on the hyperparameter of the kernel k: the standard deviation $\sigma$ in Table 2 in the revised Appendix, and copied the results in below Table R1-1. We found performance been stable with variation within 2% absolute mIoU. These results justified our system is robust.
>
> **Table R1 - 1**
>
> | Cityscapes (1024x1024 downsample to 96x96) |  |  |  |  |
> |:---:|:---:|:---:|:---:|---|
> | $\sigma$ (pixels) | 15 | 26 | 32 |  |
> | mIoU | 0.35 | 0.34 | 0.32 |  |
>
> | DeepGlobe (2448x2448 downsample to 100x100) |  |  |  |  |
> |:---:|:---:|:---:|:---:|---|
> | $\sigma$ (pixels) | 20 | 25 | 33 |  |
> | mIoU | 0.43 | 0.45 | 0.43 |  |

---

### Author Response · Authors · 2021-11-20
**The collective response to reviewers**

We thank the reviewers for their thoughtful reviews, which have undoubtedly improved the quality of our manuscript. The reviewers agreed the work is well-motivated with good presentation (R1, R2, R3, R4), comprehensive experimental evaluation and good performance (R1, R3, R4).

We are also delighted to see reviewers confirm the importance, i.e. that **low-cost segmentation is needed in many computer vision systems**, and recognise the following key contributions:

- Demonstration that previous **independently** learnt downsampling method is **suboptimal** at different locations
- Proposed **end-to-end downsampling** scheme that adapts to the memory budget according to the **pixel-level segmentation difficulty**, and proved that it works better than other available solutions
- Identifying the importance of regularisation and adding the **edge loss** that makes the joint training system work
- A **plug-and-play** component that can bring the new ideas to relevant vision tasks.

We appreciate the constructive suggestions on experiments and references. We answer specific points in separate responses to each reviewer, and summarise additional experiments and key revisions below:

#### 1. **New baseline and sensitivity analysis on hyperparameters** as listed below, as summary our method outperformed new baselines and been robust across all tested hyperparameters (with absolute mIoU variation within 3%).
- Added **new baseline** results (Talebi, H., and Milanfar, P., ICCV 2021) in revised Figure 10, (Table R2-1 in rebuttal), and Section A.2 in the revised Appendix. The new baseline is a joint-learnt method **adapting interpolated pixel value** while fixing downsampling locations, following suggestions of R2(wmVs);
- Added Table 2 in the revised Appendix, (Table R1-1 in rebuttal), for sensitivity analysis on the **standard deviation of the kernel k** in Eq.3 to better clarify question from R1(wnXt);
- Added Table 3 in the revised Appendix, (Table R2-2 in rebuttal), for sensitivity analysis on different **convolutional kernel sizes of the deformation module** answering the question on the receptive field from R2(wmVs);
- Added Table 4 in the revised Appendix, (Table R2-3/ Table R3-2 in rebuttal) sensitivity analysis on different **low-res input sizes** addressing the question from R3(EMME) and the receptive field question from R2(wmVs);
- Added Table 5 in the revised Appendix, (Table R3-1 in rebuttal) for sensitivity analysis on the **weight term balancing segmentation loss and edge-loss**, to answer the question from R3(EMME);

#### 2. Revision on **writing, discussion and references:**

1) Revised the entire methods section to:
- clarify certain terms in equations (answer questions from R1)
- clarify the relationship between the joint training system and edge-loss, and highlight they are two separate key contributions (agreed by R1, R3, R4) in answering questions from R2
- clarify why the system can learn to sample trivial locations without the proposed regularisation (question from R2);
2) Clarified relationships to following references in the related works section providing comments from R2 and R3;
- Talebi, H., and Milanfar, P. (ICCV 2021)
- Kirillov, A., et. al., (CVPR 2020)
- Huynh, C., et. al., (CVPR 2021)
3) Revised abstract and introduction to better highlight key contributions

---

### Decision · Program_Chairs · 2022-01-20

**Decision:**

Accept (Poster)

**Comment:**

Overall, this paper receives positive reviews. The reviewers find the technical novelty and contributions are significant enough for acceptance at this conference. The authors' rebuttal helps address some issues. The area chair agrees with the reviewers and recommend it be accepted at this conference.